# REVISITING LONG-TERM TIME SERIES FORECASTING: AN INVESTIGATION ON AFFINE MAPPING

## ABSTRACT

Long-term time series forecasting (LTSF) has gained significant attention in recent years. While there are various specialized designs for capturing temporal dependency, previous studies have demonstrated that a single linear layer can achieve competitive forecasting performance compared to other complex architectures. In this paper, we thoroughly investigate the intrinsic effectiveness of recent approaches and find that: 1) affine mapping in some LTSF models dominates forecasting performance across commonly utilized benchmarks; 2) affine mapping can effectively capture periodic patterns but encounter challenges when predicting non-periodic signals or time series with different periods across channels; and 3) using reversible normalization and increasing input horizon can significantly enhance the robustness of models. We provide theoretical and experimental explanations to support our findings and also discuss the limitations and future works. Our framework's code is available at https://github.com/anonymous.

## 1 INTRODUCTION

Time series forecasting has become increasingly popular in recent years due to its applicability in various fields, such as electricity forecasting (Khan et al., 2020), weather forecasting (Angryk et al., 2020), and traffic flow estimation (Chen et al., 2001). With the advances in computing resources, data volume, and model architectures, deep learning techniques, such as RNN-based (Lai et al., 2018; Sagheer & Kotb, 2019) and CNN-based models (Bai et al., 2018; Wan et al., 2019), have outperformed traditional statistical methods (Anderson et al., 1978; Ariyo et al., 2014) in terms of higher capacity and robustness.

Recently, there have been increasing interests in using Transformer-based methods to capture long-term temporal correlations in time series forecasting (Zhou et al., 2021; Wu et al., 2021; Zhou et al., 2022b; Liu et al., 2022b; Shabani et al., 2023; Zhang & Yan, 2023). These methods have demonstrated promising results through various attention mechanisms and non-autoregressive generation (NAR) techniques. However, a recent work LTSF-Linear family (Zeng et al., 2023) has shown that these Transformer-based methods may not be as effective as previously thought and found that their reported forecasting results may mainly rely on one-pass prediction compared to autoregressive generation. Instead, the LTSF-Linear, which uses only one single linear layer[1], surprisingly outperformed existing complex architectures by a large margin. Built on this work, subsequent approaches (Liu et al., 2022a; Nie et al., 2023; Li et al., 2023; Wu et al., 2023) discarded the encoder-decoder architecture and focused on developing temporal feature extractors and modeling the mapping between historical inputs and predictions. While these methods have achieved improved forecasting performance, they are still not significantly better than linear models. Additionally, they often require a large number of adjustable hyper-parameters and specific training tricks, such as normalization and channel-specific processing, which may potentially affect the fairness of comparison. Based on these observations, we raise the following questions: ① Given the marginal improvement compared to a single linear layer, are temporal feature extractors effective for long-term time series forecasting across commonly used benchmarks? ② What are the underlying mechanisms explaining the effectiveness of affine mapping in time series forecasting? and ③ What are the limits of linear models?

---

[1] In general, a linear layer is defined as an affine transformation $\mathbf{Y} = \mathbf{X}\mathbf{W}^\top + \mathbf{b}$.

In the following sections, after introducing problem definition and experimental setup, we conduct a comprehensive investigation with extensive experiments and theoretical analysis to answer these questions raised above. The main contributions of this paper are:

- We investigate the efficacy of different components in recent time series forecasting models, finding that affine mapping is critical to their forecasting performance.

- We demonstrate the effectiveness of affine mapping for learning periodicity in long-term time series forecasting tasks with both theoretical and experimental evidence. Especially, we provide a closed-form solution for linear models in forecasting periodic signals, which also achieves competitive performance in commonly used benchmarks.

- We examine the limitations of linear models when dealing with more complex datasets. Specifically, linear models generally struggle with predicting aperiodic signals or multivariate time series with varying periods. Potential remedies are provided for future study.

## 2 ARE TEMPORAL FEATURE EXTRACTORS EFFECTIVE ON LTSF?

**Problem definition.** Given a historical time series observation $\mathbf{X} = [\boldsymbol{x}_1, \boldsymbol{x}_2, \ldots, \boldsymbol{x}_n] \in \mathbb{R}^{c \times n}$ with $c$ channels and $n$ time steps, forecasting tasks aim to predict the next $m$ time steps $\mathbf{Y} = [\boldsymbol{x}_{n+1}, \boldsymbol{x}_{n+2}, \ldots, \boldsymbol{x}_{n+m}] \in \mathbb{R}^{c \times m}$ where $m$ denotes forecasting horizon. We generally use a shift window to obtain each time series instance $[\boldsymbol{x}_k, \ldots, \boldsymbol{x}_{k+n-1}, \boldsymbol{x}_{k+n}, \ldots, \boldsymbol{x}_{k+n+m-1}]$ for training and testing, where $k$ is the start timestamp in the original time series.

**Experimental setup.** Our experiments are conducted on simulated time series and several commonly used real-world datasets. For fair evaluation, we follow the same data processing protocol in Nie et al. (2023). MSE (Mean Squared Error) and MAE (Mean Absolute Error) are adopted as evaluation metrics for comparison. R-squared score is used for empirical study as it can eliminate the impact of data scale. See Appendix A for more experimental details.

**LTSF framework.** Figure 1 illustrates the general LTSF framework of recent works (Liu et al., 2022a; Zhou et al., 2022a; Chen et al., 2023; Li et al., 2023; Nie et al., 2023; Wu et al., 2023), comprising three core components: RevIN (Kim et al., 2022), a reversible normalization layer[2]; a temporal feature extractor such as attention and MLP; and a linear layer that projects the final prediction results. Given the marginal improvement compared to a single linear layer, and the potential bias caused by hyper-parameter adjustment and training tricks, we have reassessed the performance of some models on public datasets. Without loss of generality, we meticulously select four notable recent developments: PatchTST (Nie et al., 2023) (attention), MTS-Mixers (Li et al., 2023) (MLP), TimesNet (Wu et al., 2023) and SCINet (Liu et al., 2022a) (convolution). All of these methods follow this framework and have achieved state-of-the-art forecasting performance, as claimed. Figure 2 demonstrates the forecasting performance of different models on ETTh1 over 4 different prediction lengths. The baseline "RLinear" refers to a single linear projection layer with

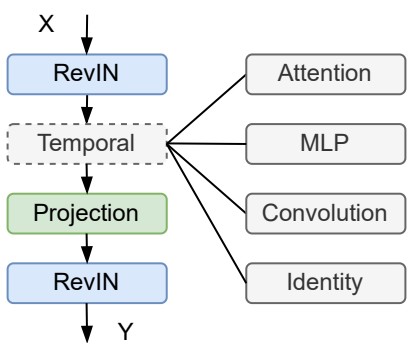

Figure 1: The general LTSF framework, comprising of RevIN (Kim et al., 2022), a temporal feature extractor, and a linear projection layer.

RevIN. The fixed random extractor means that we only randomly initialize the temporal feature extractor and do not update its parameters in the training phase. It is worth noting that the RevIN significantly improves the forecasting accuracy of these approaches. Thus, comparing one method with others which do not use RevIN may lead to unfair results due to its advantage. With the aid of RevIN, even a simple linear layer can outperform current state-of-the-art baseline PatchTST.

Remarkably, our findings suggest that even using a randomly initialized temporal feature extractor with untrained parameters can induce competitive, even better forecasting results. This intriguing

---

[2]Some works (Wu et al., 2021; Wang et al., 2023) utilize disentanglement or other normalization methods to reduce distribution shift as claimed. We will demonstrate their commonalities in the next section.

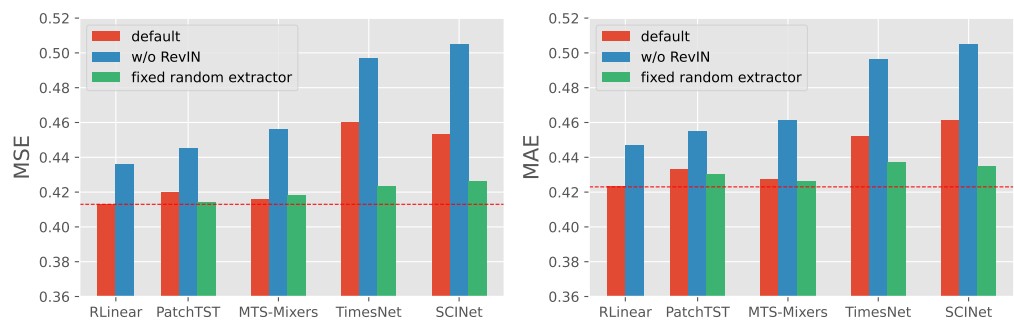

Figure 2: Forecasting results of selected models on ETTh1 (Zhou et al., 2021) dataset. MSE and MAE results are averaged from 4 different prediction lengths {96, 192, 336, 720}. The lower MSE and MAE indicate the better forecasting performance.

phenomenon raises a question of what these feature extractors have learned from time series data. Figure 3 illustrates the weights of the final linear projection layer and different temporal feature extractors, such as MLP and attention on ETTh1. Interestingly, when the temporal feature extractor is a MLP, both MLP and projection layer learn chaotic weights, whereas the product of the two remains consistent with the weights learned from a single linear layer. On the other hand, using attention as the temporal feature extractor also learns about messy weights, but the weight learned by the projection layer is similar to that of a single linear layer, implying the importance of learning a mapping pattern from input to output time series, as learned in a single linear layer, in LTSF tasks.

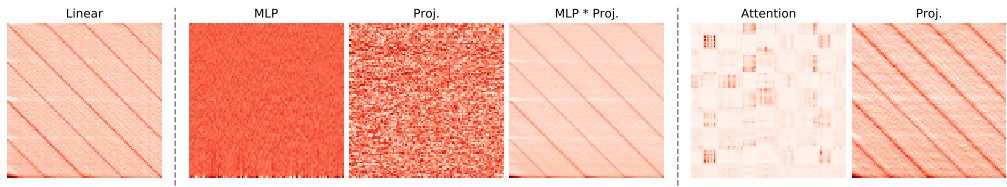

Figure 3: Weights visualization on ETTh1 where the input and output length are set as 96.

Table 1: Forecasting results on the full ETT benchmarks. The length of the historical and prediction horizon are set as 336. † indicates the temporal feature extractor with frozen random weights.

| Dataset | ETTh1 | | ETTm1 | | ETTh2 | | ETTm2 | |
|---|---|---|---|---|---|---|---|---|
| Method | MSE | MAE | MSE | MAE | MSE | MAE | MSE | MAE |
| *RLinear* | *0.420* | *0.423* | *0.370* | *0.383* | *0.325* | *0.386* | *0.273* | *0.326* |
| PatchTST | 0.431 | 0.436 | **0.366** | 0.392 | 0.331 | **0.380** | **0.276** | 0.332 |
| †PatchTST | **0.429** | **0.435** | 0.371 | **0.389** | **0.328** | 0.384 | 0.280 | **0.331** |
| MTS-Mixers | **0.414** | 0.425 | 0.378 | 0.399 | 0.353 | 0.407 | 0.291 | 0.337 |
| †MTS-Mixers | 0.423 | **0.424** | **0.377** | **0.392** | **0.351** | **0.405** | **0.282** | **0.334** |
| TimesNet | 0.493 | 0.468 | 0.406 | 0.418 | 0.358 | 0.420 | **0.304** | 0.353 |
| †TimesNet | **0.428** | **0.439** | **0.384** | **0.400** | **0.342** | **0.406** | 0.306 | **0.351** |
| SCINet | 0.467 | 0.469 | 0.404 | 0.423 | 0.365 | 0.414 | 0.329 | 0.369 |
| †SCINet | **0.428** | **0.431** | **0.386** | **0.398** | **0.349** | **0.403** | **0.299** | **0.345** |

To mitigate potential dataset-specific bias, we conducted more experiments on the full ETT benchmarks, using the same comparison protocol as (Nie et al., 2023). Table 1 demonstrates the forecasting results of RLinear and selected models. Interestingly, the simple baseline RLinear has competitive or even better performance in most cases compared to carefully designed methods. Sometimes,

these complicated models even perform worse than their prototypes with frozen randomly initialized weights in their temporal feature extractors. These intriguing observations prompt us to question whether temporal feature extractors are effective in LTSF tasks, and why a single linear layer (i.e. affine transformation) works well on commonly used benchmarks.

# 3 THEORETICAL AND EMPIRICAL STUDY OF LINEAR MODELS ON LTSF

## 3.1 ROLES OF AFFINE MAPPING IN FORECASTING

Given the input historical time series $\mathbf{X}$ and the prediction $\mathbf{Y}$, consider a single linear layer[3] as

$$\mathbf{Y} = \mathbf{X}\mathbf{W} + \mathbf{b}, \tag{1}$$

where $\mathbf{W} \in \mathbb{R}^{n \times m}$ is the weight, also termed as the transition matrix, and $\mathbf{b} \in \mathbb{R}^{1 \times m}$ is the bias.

**Assumption 1.** *A general time series $x(t)$ can be disentangled into seasonality part $s(t)$ and trend part $f(t)$ with tolerable noise (Holt, 2004; Zhang & Qi, 2005), denoted as $x(t) = s(t) + f(t) + \epsilon$.*

Numerous methods (Wu et al., 2021; Zhou et al., 2022b; Woo et al., 2022a;b; Zeng et al., 2023) have been developed to decompose time series into seasonal and trend signals, leveraging neural networks to capture periodicity and supplement trend prediction. However, apart from trend terms, it's worth noting that a single linear layer can also effectively learn periodic patterns.

**Theorem 1.** *Given a seasonal time series satisfying $x(t) = s(t) = s(t - p)$ where $p \leq n$ is the period and $t$ is the subscript index, there always exists an analytical solution for the linear model as*

$$[\boldsymbol{x}_1, \boldsymbol{x}_2, \ldots, \boldsymbol{x}_n] \cdot \mathbf{W} + \mathbf{b} = [\boldsymbol{x}_{n+1}, \boldsymbol{x}_{n+2}, \ldots, \boldsymbol{x}_{n+m}], \tag{2}$$

$$\mathbf{W}_{ij}^{(k)} = \begin{cases} 1, & \text{if } i = n - kp + (j \bmod p) \\ 0, & \text{otherwise} \end{cases}, 1 \leq k \in \mathbb{Z} \leq \lfloor n/p \rfloor, b_i = 0. \tag{3}$$

**Corollary 1.1.** *When the given time series satisfies $x(t) = ax(t - p) + c$ where $a, c$ are scaling and translation factors, the linear model still has a closed-form solution to Equation 2 as*

$$\mathbf{W}_{ij}^{(k)} = \begin{cases} a^k, & \text{if } i = n - kp + (j \bmod p) \\ 0, & \text{otherwise} \end{cases}, 1 \leq k \in \mathbb{Z} \leq \lfloor n/p \rfloor, b_i = \sum_{l=0}^{k-1} a^l \cdot c. \tag{4}$$

Equation 3 indicates that affine mapping can perfectly predict periodic signals when the length of the input historical sequence is not less than the period, but that is not a unique solution. Since the values corresponding to each timestamp in $s(t)$ are almost impossible to be linearly independent, the solution space for the parameters of $\mathbf{W}$ is extensive. In particular, it is possible to obtain a closed-form solution for more potential values of $\mathbf{W}^{(k)}$ with different factor $k$ when $n \gg p$. The linear combination of $[\mathbf{W}^{(1)}, \ldots, \mathbf{W}^{(k)}]$ with proper scaling factor also satisfies the solution of Equation 2.

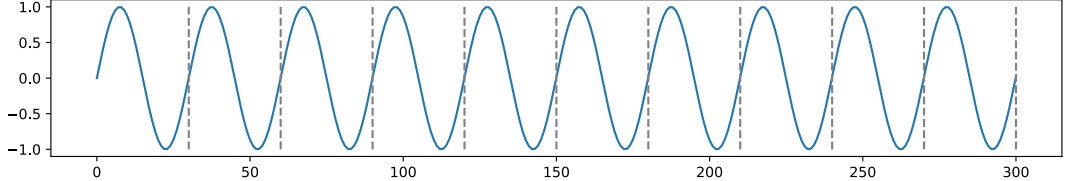

Figure 4: A simulated sine wave with a period of 30 and the length of 3000.

Consider a sine wave $y = \sin(\frac{2\pi}{30}x)$ with a period of 30 as Figure 4 shows. Let the length of input historical observation and prediction horizon be 90. According to Theorem 1, we can obtain the solution to $\mathbf{W}^{(k)}$ when $k = 1, 2, 3$ as Figure 5 illustrates. Notably, any linear combination of $[\mathbf{W}^{(1)}, \mathbf{W}^{(2)}, \mathbf{W}^{(3)}]$ still satisfies prediction conditions as long as the coefficients sum up to 1. Figure 6 illustrates the forecasting results on the simulated sine wave. Notably, this prediction method based on periodicity is insensitive to the form of the input data; it is only the period that

---

[3]We have omitted the transpose symbol here to streamline the proof process.

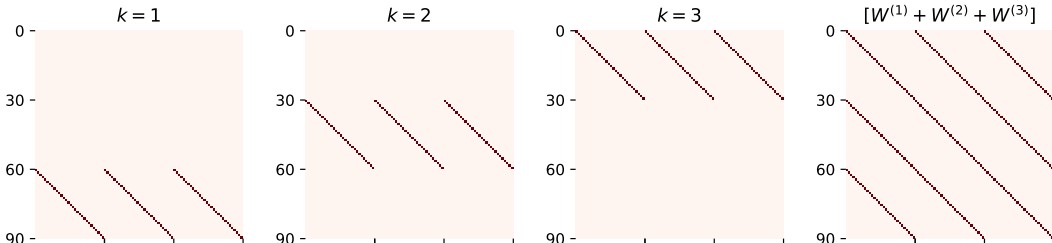

Figure 5: Visualization of the possible transition matrix $\mathbf{W}$ on the simulated sine wave.

determines the weight patterns. This may explain why a single linear layer performs well in many real-world datasets, since they often exhibit seasonality in days, weeks, months, and years. Weights visualized in Figure 3 also support our viewpoint where the transition matrix from input to output shows significant periodicity (24 time steps per period).

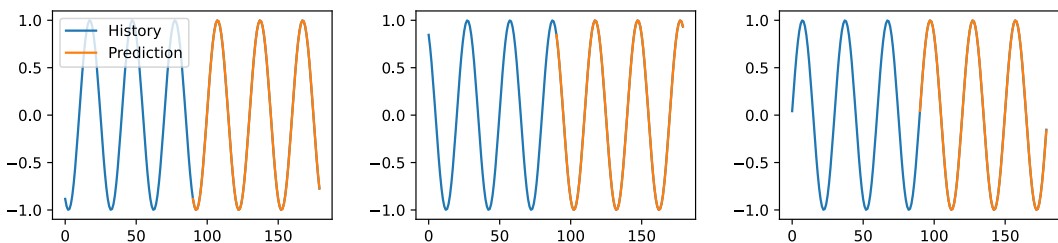

Figure 6: Forecasting showcases on the simulated sine wave.

However, in practice, time series generally follow the Assumption 1 so that the trend term may affect the learning of linear models. Figure 7 illustrates the forecasting results of a single linear layer on simulated seasonal and trend signals, including a sine wave, a linear function, and their sum. As expected, the linear model fits seasonality well but performs poorly on the trend, regardless of whether it has a bias term or not[4]. Chen et al. (2023) have also studied similar issues and provided an upper bound of linear models when forecasting time series with seasonal and trend components. Based on their work, we have adjusted their conclusion and derived the following theorem.

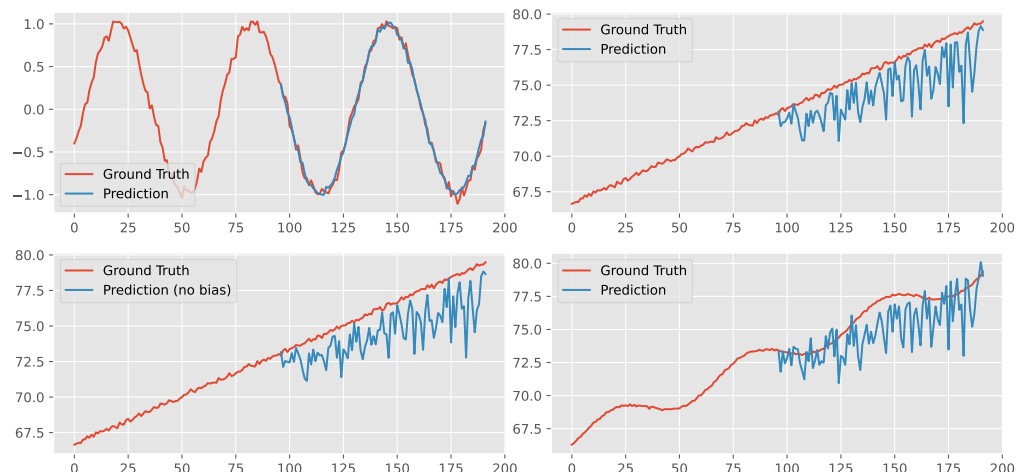

Figure 7: Forecasting visualization of a linear model on simulated seasonal and trend signals.

---

[4] $\Delta\mathbf{W}$ and $\Delta\mathbf{b}$ based on gradient descent are coupled. See Appendix C for more details

**Theorem 2.** *Let $x(t) = s(t) + f(t)$ where $s(t)$ is a seasonal signal with period $p$ and $f(t)$ satisfies $K$-Lipschitz continuous. Then there exists a linear model as Equation 2 with input horizon size $n = p + \tau, \tau \geq 0$ such that $|x(n + j) - \hat{x}(n + j)| \leq K(p + j), j = 1, \ldots, m$.*

Although the forecasting error of linear models for trend terms is bounded, it can still impact prediction results as the timestamp accumulates or the trend term becomes more significant. In prior work, intentional or unintentional strategies aimed to address this problem fall into two categories: disentanglement and normalization. We will now delve into their mechanisms and limitations.

### 3.2 Remedy for Trend: Disentanglement and Normalization

**Disentanglement.** If the trend term can be eliminated or separated from the seasonal term, forecasting performance can be improved. Previous works (Holt, 2004; Zhang & Qi, 2005; Wu et al., 2021; Woo et al., 2022a;b; Zhou et al., 2022b; Zhang et al., 2022; Zeng et al., 2023; Wang et al., 2023) have focused on disentangling time series into seasonal and trend components to predict them individually. In general, they utilized the moving average implemented by an average pooling layer with a sliding window of a proper size to get trend information from the input time series. Then, they identified seasonal features from periodic signals obtained by subtracting trend items from the original data. However, these disentanglement methods still have some problems. Firstly, the sliding window size should be larger than the maximum period of seasonality parts, or the decoupling will be inadequate. Secondly, due to the usage of the average pooling layer, alignment requires padding on both ends of the input time series, which inevitably distorts the sequence at the head and tail. Besides, even if the signals are completely disentangled, or they only have trend terms, the issue of under-fitting trend terms persists. Therefore, while disentanglement may improve forecasting performance, it still has a gap with some recent advanced models.

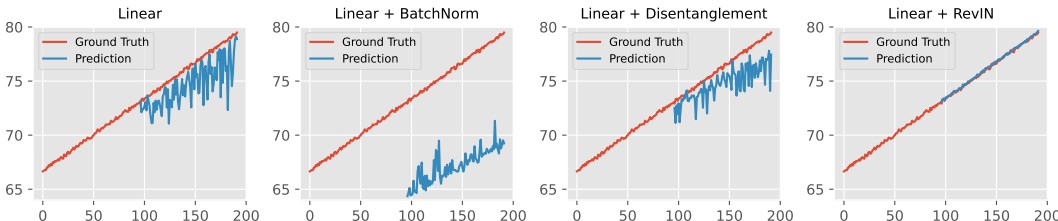

Figure 8: Forecasting results on the trend signal with different normalization methods.

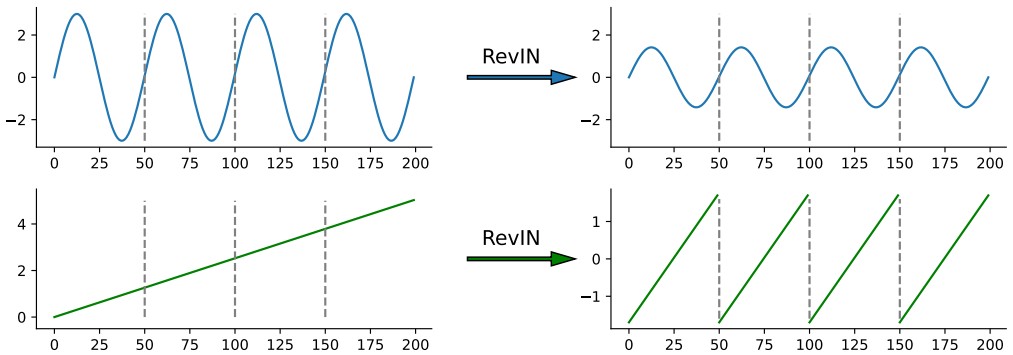

Figure 9: The effect of RevIN applied to seasonal and trend signals. Each segment separated by a dashed line contains the input historical time series $\mathbf{X}$ and the predicted sequence $\mathbf{Y}$.

**Reversible normalization.** Kim et al. (2022) recognized that some statistical information of time series, such as mean and variance, may continuously change over time. To address this challenge, they developed RevIN, a method that first normalizes the input historical time series and feeds them into forecasting modules before denormalizing the prediction for final results. Previous works (Liu et al., 2022c; Fan et al., 2023b; Wu et al., 2023; Nie et al., 2023) have attributed the effectiveness

of RevIN more to normalization for alleviating distribution shift. However, the range and size of values in time series are also meaningful in real-world scenarios. Directly applying normalization to input data may erase this statistical information and lead to poor predictions. Figure 8 illustrates the forecasting results on the simulated trend signal with two channels using different normalization methods. It is challenging to fit trend changes solely using a linear layer. Applying batch normalization even induces worse results, and layer normalization results in meaningless prediction close to zero. Disentangling the simulated time series also does not work. However, with the help of RevIN, a single linear layer can accurately predict trend terms. The core of reversible normalization lies in reversibility. It eliminates trend changes caused by moment statistics while preserving statistical information that can be used to restore final forecasting results. Figure 9 illustrates how RevIN affects seasonal and trend terms. For the seasonal signal, RevIN scales the range but does not change the periodicity. For the trend signal, RevIN scales each segment into the same range and exhibits periodic patterns. RevIN is capable of turning some trends into multiple segments with a fixed and similar trend, demonstrating periodic characteristics. As a result, errors in trend prediction caused by accumulated timesteps in the past can be alleviated, leading to more accurate forecasting results.

# 4 EXPERIMENTAL EVALUATION

In this section, we first evaluate the performance of different models on real-world datasets, and then explore the the limitations encountered when linear models confront complex scenarios.

Table 2: Time series forecasting results. The length of the historical horizon is 336 and prediction lengths are {96, 192, 336, 720}. The best results are in **bold** and the second one is underlined.

| Method | | RLinear | | RMLP | | PatchTST | | TimesNet | | DLinear | |
|---|---|---|---|---|---|---|---|---|---|---|---|
| Metric | | MSE | MAE | MSE | MAE | MSE | MAE | MSE | MAE | MSE | MAE |
| ETTh1 | 96 | **0.366** | **0.391** | 0.390 | 0.410 | 0.381 | 0.405 | 0.398 | 0.418 | 0.375 | 0.399 |
| | 192 | **0.404** | **0.412** | 0.430 | 0.432 | 0.416 | 0.423 | 0.447 | 0.449 | 0.405 | 0.416 |
| | 336 | **0.420** | **0.423** | 0.441 | 0.441 | 0.431 | 0.436 | 0.493 | 0.468 | 0.439 | 0.443 |
| | 720 | **0.442** | **0.456** | 0.506 | 0.495 | 0.450 | 0.466 | 0.518 | 0.504 | 0.472 | 0.490 |
| ETTm1 | 96 | 0.301 | **0.342** | 0.298 | 0.345 | **0.293** | 0.345 | 0.335 | 0.380 | 0.306 | 0.349 |
| | 192 | **0.335** | **0.363** | 0.344 | 0.375 | **0.335** | 0.372 | 0.358 | 0.388 | 0.339 | 0.368 |
| | 336 | 0.370 | **0.383** | 0.390 | 0.410 | **0.366** | 0.392 | 0.406 | 0.418 | 0.374 | 0.390 |
| | 720 | 0.425 | **0.414** | 0.445 | 0.441 | **0.420** | 0.424 | 0.449 | 0.443 | 0.428 | 0.423 |
| ETTh2 | 96 | **0.262** | **0.331** | 0.288 | 0.352 | 0.276 | 0.337 | 0.348 | 0.392 | 0.289 | 0.353 |
| | 192 | **0.319** | **0.374** | 0.343 | 0.387 | 0.339 | 0.379 | 0.362 | 0.404 | 0.383 | 0.418 |
| | 336 | **0.325** | 0.386 | 0.353 | 0.402 | 0.331 | **0.380** | 0.358 | 0.420 | 0.448 | 0.465 |
| | 720 | **0.372** | **0.421** | 0.410 | 0.440 | 0.379 | 0.422 | 0.442 | 0.463 | 0.605 | 0.551 |
| ETTm2 | 96 | **0.164** | **0.253** | 0.174 | 0.259 | 0.165 | 0.256 | 0.188 | 0.267 | 0.167 | 0.260 |
| | 192 | **0.219** | **0.290** | 0.236 | 0.303 | 0.238 | 0.305 | 0.252 | 0.308 | 0.224 | 0.303 |
| | 336 | **0.273** | **0.326** | 0.291 | 0.338 | 0.276 | 0.332 | 0.304 | 0.353 | 0.281 | 0.342 |
| | 720 | **0.366** | **0.385** | 0.371 | 0.391 | 0.369 | 0.391 | 0.405 | 0.409 | 0.397 | 0.421 |
| Weather | 96 | 0.175 | 0.225 | **0.149** | **0.202** | 0.155 | 0.205 | 0.172 | 0.220 | 0.176 | 0.237 |
| | 192 | 0.218 | 0.260 | **0.194** | **0.242** | 0.199 | 0.245 | 0.219 | 0.261 | 0.220 | 0.282 |
| | 336 | 0.265 | 0.294 | **0.243** | **0.282** | 0.249 | 0.284 | 0.280 | 0.306 | 0.265 | 0.319 |
| | 720 | 0.329 | 0.339 | **0.316** | **0.333** | 0.319 | 0.335 | 0.365 | 0.359 | 0.326 | 0.363 |
| ECL | 96 | 0.140 | 0.235 | **0.129** | **0.224** | 0.133 | 0.226 | 0.168 | 0.272 | 0.140 | 0.237 |
| | 192 | 0.154 | 0.248 | **0.147** | **0.240** | 0.149 | 0.242 | 0.184 | 0.289 | 0.153 | 0.249 |
| | 336 | 0.171 | 0.264 | **0.164** | **0.257** | 0.167 | 0.260 | 0.198 | 0.300 | 0.169 | 0.267 |
| | 720 | 0.209 | 0.297 | **0.203** | **0.291** | 0.205 | 0.293 | 0.220 | 0.320 | 0.210 | 0.310 |

**Comparison on real-world datasets.** We perform experiments using three latest competitive baselines: PatchTST (Nie et al., 2023), TimesNet (Wu et al., 2023), and DLinear (Zeng et al., 2023). Given that RevIN significantly improves the forecasting performance, we add two simple baselines, RLinear and RMLP with two linear layers and a ReLU activation, for a fairer comparison. Table 2 provides an overview of forecasting results for all benchmarks. However, the two well-designed large models, PatchTST and TimesNet, do not exhibit a significant improvement over simple linear baselines. It is probable that the effectiveness of these models can be attributed to their ability to learn periodicity through affine mapping and the efficiency of reversible normalization. The com-

parison between RLinear and DLinear also highlights the limitations in disentanglement. Notably, we have observed that RLinear fails to outperform complex models on datasets with a substantial number of channels, such as Weather and ECL, which will be studied in the following section.

Table 3: Forecasting results on Weather and ECL of RLinear-CI where the input horizon is 336.

| Dataset | | Weather | | | | ECL | | | |
|---|---|---|---|---|---|---|---|---|---|
| Method | Metric | 96 | 192 | 336 | 720 | 96 | 192 | 336 | 720 |
| RLinear | MSE | 0.175 | 0.218 | 0.265 | 0.329 | 0.140 | 0.154 | 0.171 | 0.209 |
| | MAE | 0.225 | 0.260 | 0.294 | 0.339 | 0.235 | 0.248 | **0.264** | 0.297 |
| RLinear-CI | MSE | **0.146** | **0.189** | **0.241** | **0.314** | **0.134** | **0.149** | **0.166** | **0.202** |
| | MAE | **0.194** | **0.235** | **0.275** | **0.327** | **0.232** | **0.246** | 0.265 | **0.293** |

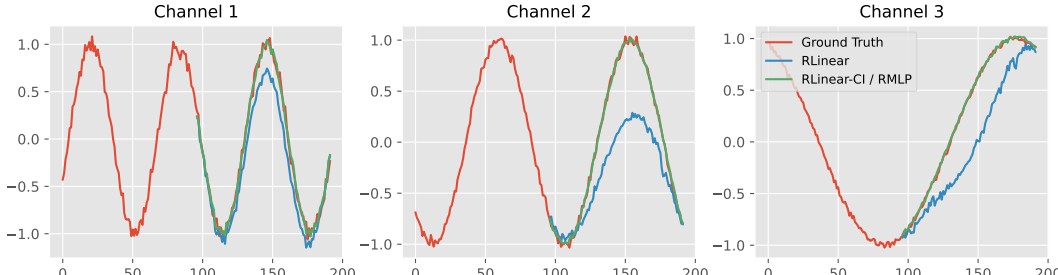

Figure 10: Forecasting results on simulated time series with three channels of different periods.

**When linear models meet multiple periods among channels.** Although affine mapping is capable of learning periodicity in time series, it faces challenges when dealing with multi-channel datasets. A viable compromise solution is to model each channel independently using a linear layer, referred to as Channel Independent (CI) modeling. Figure 10 illustrates the forecasting results of different models applied to simulated time series with three distinct periodic channels. It is observed that RLinear-CI and RMLP are able to fit curves, while RLinear fails. This suggests that a single linear layer may struggle to learn different periods within channels. Nonlinear units or CI modeling may be useful in enhancing the robustness of the model for multivariate time series with different periodic channels. Table 3 provides forecasting results on Weather and ECL of RLinear using CI, which achieves comparable performance with RMLP, confirming that a single linear layer may be vulnerable to varying periods among channels. To further investigate the effect of affine mapping on multivariate time series, we conduct simulations using a series of sine waves with angular frequencies ranging from $1/30$ to $1/3$ and the length of 3000, as Figure 11 shows.

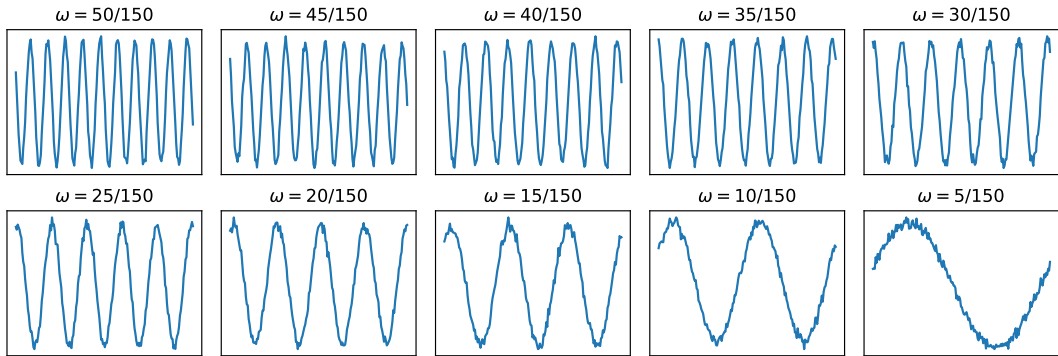

Figure 11: Simulated sine waves with angular frequency ranges from 1/30 to 1/3, and 200 in length.

Figure 12 demonstrates the forecasting results under different settings. Our findings indicate that the linear model consistently performs well on time series with two channels, regardless of whether the difference in periodicity is small or large. However, as the number of channels with different

periods increases, the linear model gradually performs worse, while models with nonlinear units or CI continue to perform well. Additionally, increasing the input horizon can effectively alleviate the forecasting performance of the linear model on multi-channel datasets. These observations suggest that existing models may focus on learning seasonality, and that the differences in periodicity among different channels in multivariate time series are key factors that constrain forecasting performance. Theorem 3 provides an explanation of linear models in forecasting multivariate time series.

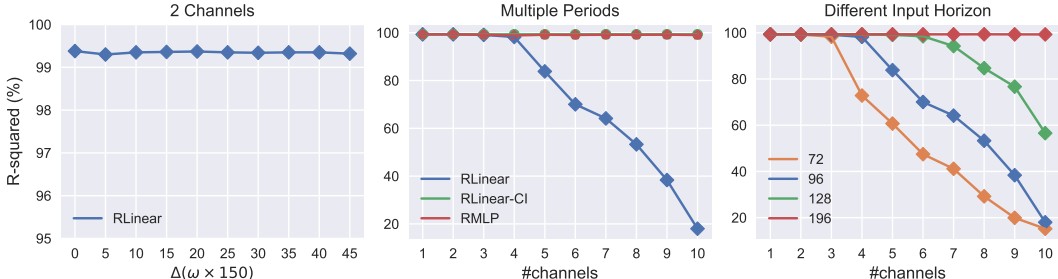

Figure 12: **Left**: Forecasting resutls on simulated 2-variate time series. $\Delta\omega$ denotes the difference of angular frequency between channels. **Middle**: Forecasting results on simulated datasets with different periodic channels. **Right**: Impact of input horizon on forecasting performance.

**Theorem 3.** *Let $\mathbf{X} = [\boldsymbol{s}_1, \boldsymbol{s}_2, \ldots, \boldsymbol{s}_c]^\top \in \mathbb{R}^{c \times n}$ be the input historical multivariate time series with $c$ channels and the length of $n$. If each signal $\boldsymbol{s}_i$ has a corresponding period $p_i$, there must be a linear model $\mathbf{Y} = \mathbf{X}\mathbf{W} + \mathbf{b}$ that can predict the next $m$ time steps when $n \geq lcm(p_1, p_2 \ldots, p_c)$.*

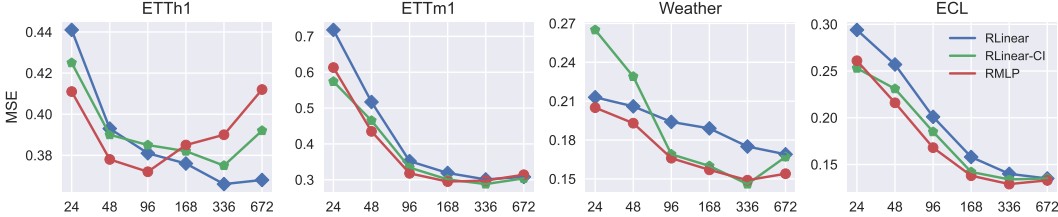

Figure 13: Impact of input horizon on forecasting results. Lower MSE indicates better performance.

As shown in Figure 13, increasing the input horizon can lead to a significant improvement in forecasting performance. This is because a longer input horizon covers more potential periods, minimizing the performance gap between linear models and those with nonlinear units. However, it is worth noting that RLinear-CI and RMLP perform worse on the ETTh1 dataset when the input horizon is longer, which may be due to the small volume of this particular dataset. Furthermore, it should be noted that there is an upper limit to the performance improvement achieved by increasing the input horizon. This limit may be highly dependent on the periodic patterns present in datasets.

## 5 CONCLUSION

This paper investigates the effect of affine mapping in LTSF, with the following important findings: (1) affine mapping in some LTSF models dominates forecasting performance across commonly utilized benchmarks, which generally prone to learn similar transition matrix from input historical observation to output prediction; 2) affine mapping can effectively capture periodic patterns, but it encounters challenges when predicting non-periodic signals or time series with different periods across channels. Furthermore, employing reversible normalization and increasing the input horizon can significantly enhance the model's robustness. We provide theoretical explanations with extensive experiments on both simulated and real-world datasets to support our findings.

**Limitations and future work.** Long-term time series benchmarks often display consistent seasonal patterns. It is worthwhile to study how models perform when the seasonality changes. It would also be valuable to explore the applicability of our theories to other tasks, such as short-term time series forecasting. We acknowledge that these explorations will be left for future work.

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

## A  EXPERIMENTAL DETAILS

**Real-world datasets.** Our experiments involve six public real-world datasets: (1) ETT (Zhou et al., 2021) (Electricity Transformer Temperature) with four datasets with different granularity records six power load features and oil temperature from electricity transformers; (2) Weather[5] contains 21 meteorological indicators in the 2020 year from nearly 1600 locations in the U.S.; and (3) ECL[6] records the hourly electricity consumption of 321 customers from 2012 to 2014. Table 4 provides detailed statistical information of all datasets. For a fair comparison, we follow the same evaluation protocol in Nie et al. (2023) and split all datasets into training, validation, and test sets. Our proposed baselines are trained using the L2 loss and the Adam (Loshchilov & Hutter, 2019) optimizer. The training process is early stopped within 20 epochs. MSE (Mean Squared Error) and MAE (Mean Absolute Error) are adopted as evaluation metrics for comparison. R-squared score is used for empirical study as it can eliminate the impact of data scale. All the models are implemented in PyTorch (Paszke et al., 2019) and tested on a single Nvidia V100 32GB GPU for three times.

Table 4: Statistical information of all datasets for time series forecasting.

| Dataset | ETTh1/h2 | ETTm1/m2 | Weather | ECL |
|---|---|---|---|---|
| #Channel | 7 | 7 | 21 | 321 |
| Timesteps | 17,420 | 69,680 | 52,696 | 26,304 |
| Granularity | 1 hour | 15 minutes | 10 minutes | 1 hour |
| Data Partition | 6:2:2 (month) | | 7:2:1 | |

**Reproduction.** We implemented the reversible normalization module using the default setting from RevIN (Kim et al., 2022). The baseline RLinear consists of RevIN and a single linear layer responsible for mapping the input observations to the output prediction, as shown in Algorithm 1. The RMLP model comprises RevIN, a MLP which includes two linear layers with ReLU activation where we set the number of hidden states to 512, and a linear projection layer. The baseline RLinear-CI involves $c$ linear layers for modeling $c$ channels in multivariate time series individually.

---

**Algorithm 1** PyTorch-like RLinear Code

```python
import torch.nn as nn
import RevIN

class RLinear(nn.Module):
    """
    Input: tensor in shape [batch_size, sequence_length, #channel].
    """
    def __init__(self, input_size, output_size, channel):
        super().__init__()
        self.linear = nn.Linear(input_size, output_size)
        self.revin = RevIN(channel)

    def forward(self, x):
        # mean, std = x.mean(), x.std()
        x = self.revin(x, 'norm') # similar as (x - mean) / std
        x = self.linear(x.transpose(1, 2)).transpose(1, 2)
        x = self.revin(x, 'denorm') # similar as x * std + mean

        return x
```

---

**Details on benchmarks and baselines.** We adopt the same pre-processing protocol in (Nie et al., 2023). Across all benchmarks, we set the initial learning rate to 0.005 and the batch size to 128. The results of PatchTST, MTS-Mixers, TimesNet, SCINet, and DLinear are based on our reproduction. For models with a fixed random temporal feature extractor, we initialize these models with the fixed random seed 1024. We adopt the hyper-parameters suggested in their original papers for each model.

---

[5]https://www.ncei.noaa.gov/data/local-climatological-data/
[6]https://archive.ics.uci.edu/ml/datasets/ElectricityLoadDiagrams20112014

## B    PROOFS

For better readability, we have re-listed the unproven theorems as follows.

**Theorem B.1.** *Given a seasonal time series satisfying $x(t) = s(t) = s(t - p)$ where $p \leq n$ is the period and $t$ is the subscript index, there always exists an analytical solution for the linear model as*

$$[\boldsymbol{x}_1, \boldsymbol{x}_2, \ldots, \boldsymbol{x}_n] \cdot \mathbf{W} + \mathbf{b} = [\boldsymbol{x}_{n+1}, \boldsymbol{x}_{n+2}, \ldots, \boldsymbol{x}_{n+m}], \tag{B.1}$$

$$\mathbf{W}_{ij}^{(k)} = \begin{cases} 1, & \textit{if } i = n - kp + (j \bmod p) \\ 0, & \textit{otherwise} \end{cases}, 1 \leq k \in \mathbb{Z} \leq \lfloor n/p \rfloor, b_i = 0. \tag{B.2}$$

*Proof.* $\forall \boldsymbol{x}_{n+j}, 1 \leq j \leq m, z \in \mathbb{Z}^+, \boldsymbol{x}_{n+j} = \boldsymbol{x}_{n-zp+(j \bmod p)}$ according to $x(t) = x(t-p)$, thus we have $[\boldsymbol{x}_1, \boldsymbol{x}_2, \ldots, \boldsymbol{x}_n] \cdot \mathbf{W}^{(k)} = [\boldsymbol{x}_{n-kp+1}, \boldsymbol{x}_{n-kp+2}, \ldots, \boldsymbol{x}_{n-kp+m}] = [\boldsymbol{x}_{n+1}, \boldsymbol{x}_{n+2}, \ldots, \boldsymbol{x}_{n+m}].$ □

**Corollary B.1.1.** *When the given time series satisfies $x(t) = ax(t - p) + c$ where $a, c$ are scaling and translation factors, the linear model still has a closed-form solution to Equation B.1 as*

$$\mathbf{W}_{ij}^{(k)} = \begin{cases} a^k, & \textit{if } i = n - kp + (j \bmod p) \\ 0, & \textit{otherwise} \end{cases}, 1 \leq k \in \mathbb{Z} \leq \lfloor n/p \rfloor, b_i = \sum_{l=0}^{k-1} a^l \cdot c. \tag{B.3}$$

*Proof.* $\forall \boldsymbol{x}_{n+j}, 1 \leq j \leq m, z \in \mathbb{Z}^+, \boldsymbol{x}_{n+j} = a^z \boldsymbol{x}_{n-zp+(j \bmod p)} + \sum_{l=0}^{z-1} a^l \cdot c$ according to $x(t) = ax(t-p)+c$, thus we have $[\boldsymbol{x}_1, \ldots, \boldsymbol{x}_n] \cdot \mathbf{W}^{(k)} + \mathbf{b} = [a^k \boldsymbol{x}_{n-kp+1} + \sum_{l=0}^{k-1} a^l \cdot c, \ldots, a^k \boldsymbol{x}_{n-kp+m} + \sum_{l=0}^{k-1} a^l \cdot c] = [\boldsymbol{x}_{n+1}, \ldots, \boldsymbol{x}_{n+m}].$ □

**Theorem B.2.** *Let $x(t) = s(t) + f(t)$ where $s(t)$ is a seasonal signal with period $p$ and $f(t)$ satisfies K-Lipschitz continuous. Then there exists a linear model as Equation 2 with input horizon size $n = p + \tau, \tau \geq 0$ such that $|x(n + j) - \hat{x}(n + j)| \leq K(p + j), j = 1, \ldots, m$.*

*Proof.* To simplify the proof process, we assume that the timestamp of historical data is 1 to $n$. Then for the j-th true value $x(n + j)$ to be predicted, we have

$$x(n + j) = x(p + \tau + j) = s(\tau + j) + f(p + \tau + j). \tag{B.4}$$

Supposing that the linear model can only learn periodic patterns, we can directly use Equation 3 as an approximate solution where we choose $k = 1$. Thus, the prediction for $x(n + j)$ is

$$\hat{x}(n + j) = \mathbf{XW} + \mathbf{b} = x(n - p + (j \bmod p)) = s(\tau + j) + f(\tau + (j \bmod p)). \tag{B.5}$$

Leveraging properties of K-Lipschitz continuous we can get

$$\begin{aligned} |x(n + j) - \hat{x}(n + j)| &= |f(p + \tau + j) - f(\tau + (j \bmod p))| \\ &\leq K|p + j - (j \bmod p)| \\ &\leq K(p + j). \end{aligned} \tag{B.6}$$

□

**Theorem B.3.** *Let $\mathbf{X} = [\boldsymbol{s}_1, \boldsymbol{s}_2, \ldots, \boldsymbol{s}_c]^\top \in \mathbb{R}^{c \times n}$ be the input historical multivariate time series with $c$ channels and the length of $n$. If each signal $\boldsymbol{s}_i$ has a corresponding period $p_i$, there must be a linear model $\mathbf{Y} = \mathbf{XW} + \mathbf{b}$ that can predict the next $m$ time steps when $n \geq lcm(p_1, p_2 \ldots, p_c)$.*

*Proof.* Apparently $p = lcm(p_1, p_2 \ldots, p_c)$ is the least common period for all channels. According to Equation B.2, there must be a linear model satisfying Equation B.1 for each channel $\boldsymbol{s}_c \in \mathbb{R}^{1 \times n}$. □

| RLinear | RLinear-CI | RLinear-CI | RLinear-CI |

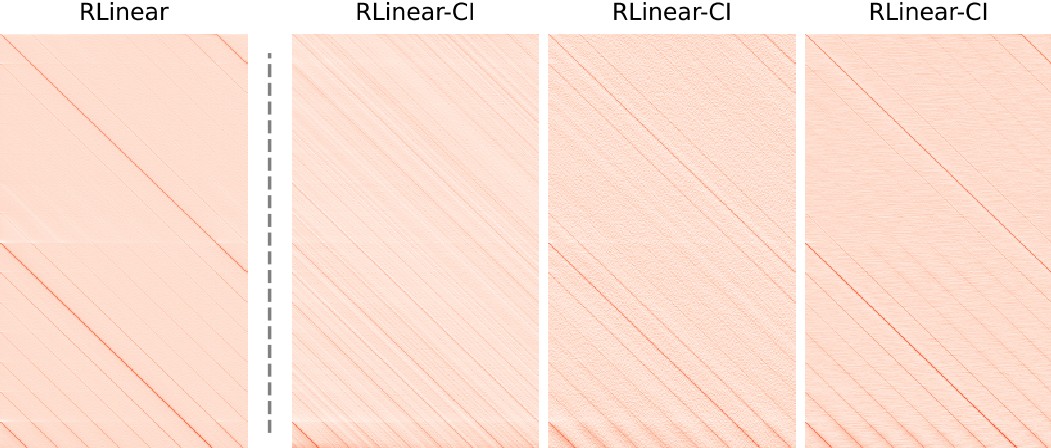

Figure 14: Visualizations of $\mathbf{W}$ of RLinear and RLinear-CI on ECL dataset.

## C  DEFECTS OF LINEAR MODELS

Although Theorem 3 provides a solution for predicting multivariate time series with different periodic channels, an excessively long input horizon may make the optimization of $\mathbf{W}$ difficult due to its sparsity. Figure 14 shows different $\mathbf{W}$ in RLinear and three linear layers of RLinear-CI where the length of input horizon is 336 and prediction horizon is 192. Apparently, with the help of CI, different projection layers have learned different periodic features, which are more detailed. A single linear layer focuses more on global periodicity, but prone to forget fine-grained periodic features. The study of how to balance periodicity at different scales requires further investigation.

Besides, the optimization process for the weight and bias parameters in linear models can potentially limit their ability to accurately fit the time series data, as shown in Equation C.7 where $\mathbf{W}$ is the weight, $\mathbf{b}$ is the bias, $\eta$ denotes the learning rate, and $L$ represents the loss function. Due to the common term $\partial L/\partial \mathbf{Y}$, the update of $\mathbf{W}$ and $\mathbf{b}$ is coupled. This coupling is what allows the network to adjust both weight and bias in a coordinated way to reduce the loss. However, in certain scenarios, such as fitting trend signals, it may be necessary for the model to learn a substantial bias, while maintaining the weight as an identity transition matrix. This could provide insight into why linear models tend to underperform when applied to trend signals.

$$
\begin{aligned}
\mathbf{W} &= \mathbf{W} - \eta\frac{\partial L}{\partial \mathbf{W}} = \mathbf{W} - \eta\mathbf{X}^{\top}\frac{\partial L}{\partial \mathbf{Y}}, \\
\mathbf{b} &= \mathbf{b} - \eta\frac{\partial L}{\partial \mathbf{b}} = \mathbf{b} - \eta\frac{\partial L}{\partial \mathbf{Y}}.
\end{aligned}
\tag{C.7}
$$

## D  NONLINEAR UNITS WITH MULTIVARIATE TIME SERIES

In our main experiment and subsequent investigations, we discovered that models utilizing nonlinear units or the aid of CI projections exhibit superior performance when handling signals with various periodic channels. The efficacy of CI is easily comprehensible according to Theorem 3. As for the other one, some concurrent works (Fan et al., 2023a; Villani & McBurney, 2023) posit that nonlinear units such as ReLU can be decomposed into a set of linear models, each defined within a partitioned region of the input space. This could explain why models with nonlinear units perform well when processing time series with numerous distinct periodic channels.

