# OpenReview forum: "Revisiting Long-term Time Series Forecasting: An Investigation on Affine Mapping"
_ICLR.cc/2024/Conference — Submitted to ICLR 2024_

### Official Review · Reviewer_TkUX · 2023-10-29

**Soundness:** 3 good
**Presentation:** 3 good
**Contribution:** 2 fair
**Rating:** 6
**Confidence:** 4

**Summary:**

This paper offers a timely and rigorous critique of the recent trend of using transformer-based models for long-term time series forecasting, and arrives at several surprising and important empirical conclusions. One of these conclusions is that just a simple linear layer (as in DLinear) combined with RevIN (reversible instance normalization with a learnable affine mapping of statistics) results in superior long-term forecast performance. The second is, even more shockingly, that many popular transformer-based methods with random parameter initializations already outperform their trained counterparts on long-term multivariate point forecasting tasks. The paper further investigates failure modes of single-layer linear forecasters in the multivariate setting.

The paper is well-written and of interest to the community in terms of the findings it offers. However, the novelty of the paper is limited for both methodology and theory.

**Strengths:**

The paper is well-written, and its empirical findings are surprising and insightful for time series forecasting practitioners.

The first such conclusion results in a new and easy-to-implement model: RLinear. RLinear is simply DLinear with RevIN, and offers competitive performance. The second empirical exploration calls into question the use of large transformer architectures for forecasting. When using transformers to "extract features" the authors find that the learned mappings are easily confused and fail to capture the most basic signals with periodicity. In fact, this effect of "overfitting" appears so pronounced that the random initializations of these models appear to generalize almost better than their fitted counterparts. This is a very suprising finding that I look forward to verifying with the experimental setup the authors will make available.

The paper then moves to discussing why linear maps suffice to fit periodic signals. They offer an interesting comparative study of how linear models behave with different methods for normalization/disentanglement.

**Weaknesses:**

My main critique of the paper is its limited novelty. While the insights offered by the paper are quite useful, the main methodological advance offered simply combines two very recent ideas. Moreover, the validity of this method is also not rigorously tested with an ablation/comparative study. For example, there is no study as in Table 2 for the idea in Figure 8.

The theoretical results proposed in the paper are hardly novel. For example, Thm 1 barely needs to be denoted so (periodic functions can be reconstructed by shifting, and shifting is a linear operator..). Similarly, Thm 3 produces a somewhat unsurprising conclusion which is not contextualized well in the paper's presentation.

Among other points the authors may like to consider:
- For the uninitiated reader, the architecture of RLinear (and its simplicity) are hard to understand, and Figure 1 are hard to understand. Perhaps they could be revisited
- Footnote 4 and how it relates to the context is unclear.
- Please cite the DLinear paper with the accepted venue.
- Please revisit the conclusion section for grammar: e.g.,  "investigate" "where they generally prone" "encounter"
- Just a suggestion: The recent TiDE paper attacks some of the same problems as the authors. Although that work does not appear to be peer-reviewed yet it may be worthwihle to briefly discuss it in context.

**Questions:**

N/A

---

> ### Author Response · Authors · 2023-11-23
> **Response to Reviewer TkUX (1/1)**
>
> Thank you for your thoughtful review and for highlighting the strengths of our work. Your critiques are greatly appreciated and we believe they will significantly improve the quality of our paper. We would like to address your concerns as follows:
>
> > **C1: The limited novelty of the study and theoretical results**
>
> We understand your perspective on the novelty of the study and theoretical results because they seems trivial and hardly novel. However, it is important to clarify that our primary objective in this study was not to introduce new algorithms, but rather to illuminate the intrinsic efficacy of affine mapping in LTSF tasks. We believe that our theoretical analysis and empirical studies do represent a substantial technical contribution. We not only elucidate why linear models perform well in LTSF tasks, but also remind future researchers to conduct more ablation experiments as we did in Section 2 to confirm the effectiveness of their methods. Additionally, we highlight the impact of the input horizon length and tricks like RevIN. This findings could potentially guide future research and enhance the development of more effective models.
>
> > **C2: The rigour of the comparative study in Table 2 for the idea in Figure 8.**
>
> We acknowledge that a more rigorous ablation study would strengthen our paper. However, the primary aim of Table 2 is to show that simple baselines, such as RLinear and RMLP, can yield competitive performance compared to recent complicated methods on LTSF tasks, implying potential questions regarding the effectiveness of these methods and the suitability of the datasets used. On the other hand, Figure 8 illustrates the limitations of linear models in fitting trend terms, both with and without the help of disentanglement and normalization. The purposes served by the table and the figure are distinct and complementary, with each contributing unique insights to our analysis.
>
> > **C3: Clearer explanation and clarification of some points**
>
> Thank you for your thoughtful suggestions to our presentation. We will revise Figure 1 and provide more detailed explanations of used baselines in Appendix for the readers. As for Footnote 4, we can write the full update rule of $W,b$ as $W=W-\eta\frac{\partial L}{\partial W}=W-\eta X^\top\frac{\partial L}{\partial Y}$ and $b=b-\eta\frac{\partial L}{\partial b}=b-\eta\frac{\partial L}{\partial Y}$ where $L$ denotes the loss function. Apparently the update of the weight and bias is coupled due to $\frac{\partial L}{\partial Y}$. We will provide additional explanations to clarify its importance in the context of our discussion. As for citing the DLinear paper formally, we thank you for pointing out the omission. We will update the citation with the accepted venue for the DLinear paper. As for grammar issues in the conclusion section, we will revise the conclusion section to correct these issues. As for discussing the TiDE paper, we are aware of the TiDE paper and agree that it tackles similar problems to those addressed in our paper. We will include a discussion on this paper in our revised manuscript, acknowledging its relevance to our work.
>
> We hope that these responses address your concerns and that our revisions will make the paper clearer and more valuable. Thank you again for your thoughtful feedback.

---

### Official Review · Reviewer_SH4h · 2023-10-30

**Soundness:** 3 good
**Presentation:** 3 good
**Contribution:** 3 good
**Rating:** 6
**Confidence:** 3

**Summary:**

The authors of this work study the long-term time series forecasting (LTSF) problem. They demonstrate that a single linear layer can perform competitively for LTSF compared to other complex architectures. Specifically, with theoretical analysis and empirical studies, the authors find that affine mapping is critical in LTSF tasks and inspect its efficacy. While dealing with multivariate time series data, the limitation of linear models is also discussed.

**Strengths:**

1. The authors study the fundamental mechanisms that affect the performance of recent LTSF models, which is important for the community.
2. This work provides theoretical and empirical evidence to support its findings.
3. The presentation is well-organized and easy to follow.

**Weaknesses:**

1. As a research work submitted to a top-tier ML conference, the technical contribution is limited. In particular, novel solutions/algorithms that leverage the important findings are expected.
2. It is better to investigate more real-world time series datasets with perplexing patterns.

**Questions:**

What about the efficacy of affine mapping when the models deal with noisy and low-quality time series data?

---

> ### Author Response · Authors · 2023-11-23
> **Response to Reviewer SH4h (1/1)**
>
> Thank you for your valuable feedback and insights. We appreciate your recognition of the importance of our study and its contribution to the understanding of the fundamental mechanisms affecting the performance of recent LTSF models. We would like to address your concerns as follows.
>
> > **C1: Limited technical contribution**
>
> We appreciate your feedback regarding the perceived absence of innovative solutions or algorithms in our paper. However, it is important to clarify that our primary objective in this study was not to introduce new algorithms, but rather to illuminate the intrinsic efficacy of affine mapping in LTSF tasks. We believe that our theoretical analysis and empirical studies do represent a substantial technical contribution. We not only elucidate why linear models perform well in LTSF tasks, but also remind future researchers to conduct more ablation experiments as we did in Section 2 to confirm the effectiveness of their methods. Additionally, we highlight the impact of the input horizon length and tricks like RevIN. This findings could potentially guide future research and enhance the development of more effective models. We are looking forward to see new algorithms that leverage these insights, such as how to cope with trend terms without RevIN, and how to process multi-channel datasets as mentioned in Section 4.
>
> > **C2: Need for more real-world datasets**
>
> We agree with your suggestion that testing on more diverse real-world datasets could further strengthen our results. However, it is crucial to note that the datasets we currently utilize for evaluation are also widely used in other LTSF benchmarks such as Informer, FEDformer, DLinear, and PatchTST. Given the setting of LTSF tasks which covers longer input horizon and prediction length, these tasks inherently contain increased seasonality. Thus, we are prompted to question the rationale behind LTSF tasks, as they tend to heavily rely on inherent seasonality within datasets. This raises potential concerns about the datasets currently in widespread use for evaluation purposes. It is possible that these datasets, with their inherent biases, may not provide a comprehensive or fair assessment of different forecasting methodologies. As we stated in Section 3.1, this may explain why a single linear layer performs well in many real-world datasets, since they often exhibit seasonality in days, weeks, months, and years. We also list one "abnormal" dataset in C3 to further support our viewpoints.
>
> > **C3: What about the efficacy of affine mapping when the models deal with noisy and low-quality time series data?**
>
> Indeed, our current datasets are noisy and of low quality. However, in LTSF tasks, the introduction of a longer observation window allows for the incorporation of more seasonal features, which in turn helps mitigate the impact of these noises. Here we introduce one "abnormal" dataset, Exchange, which has been evaluated in DLinear and PatchTST. This dataset comprises changes in exchange rates across eight countries over several years and exhibits minimal periodic characteristics. The table below presents the forecasting results from both DLinear and PatchTST.
>
> |Method|DLinear||PatchTST||Repeat|&emsp;
> |:-:|:-:|:-:|:-:|:-:|:-:|:-:
> |H|MSE|MAE|MSE|MAE|MSE|MAE
> |96|**0.081**|0.203|0.082|0.199|**0.081**|**0.196**
> |336|**0.305**|0.414|0.319|0.407|**0.305**|**0.396**
>
> Although the MSE and MAE evaluation metrics appear low, the actual visualized prediction results are disappointing. DLinear discovered that simply repeating the final value of historical time series outperforms all the baselines, while PatchTST deemed 'Exchange' as "chaotic" and consequently excluded it from their evaluation. We believe that this instance further supports the viewpoints we expressed in C2.
>
> ---
>
> Once again, we appreciate your thoughtful comments and questions. They have provided us with valuable directions for improving our work and for future research.

---

### Official Review · Reviewer_uhGz · 2023-10-30

**Soundness:** 1 poor
**Presentation:** 3 good
**Contribution:** 1 poor
**Rating:** 3
**Confidence:** 4

**Summary:**

This paper investigates the recent approaches in Long-Term Time Series Forecasting (LTSF) especially around the one-pass prediction over a single linear layer that is seen to achieve competitive performance on par with other complex architectures.
The authors state that affine mapping in LTSF models effectively capture periodic patterns and thus dominates in the forecasting performance over the baseline models. However, there are no information about what affine mapping is or how affine mapping is effectively introduced in the LTSF models to improve performance.
The authors state that a single linear projection layer with Reversible Normalization (RevIN) outperforms the state-of-the-art models currently because the single layer feature extractor learns weights that are consistent with the projection layer which indeed imply a mapping pattern between input to output series. But this doesn’t effectively prove, affine mapping is the reason for the performance improvement.
The authors investigate the disentanglement of seasonal and trend terms of the time series model and state that there is research gap on the advanced models still left to explore. The authors also question the effectiveness of temporal feature extractors on LTSF tasks but there is no investigation around it that leads to the impact of the temporal feature extractors on the model performance other than affine mapping.
Finally, the authors run a series of experimental evaluation and state that the large models, PatchTST and TimesNet do not exhibit significant improvement over baseline single layer models and the authors are assuming it to be the models’ efficacy to learn periodicity through affine mapping. Also, in multiple channel use-cases, a single layer is observed to fail without modelling each channel independently which again questions the findings of the investigation.

**Strengths:**

1. The paper gives a detailed analysis of various use-cases of LTSF on public datasets and makes solid observations.
2. The authors prove the importance of Reversible Normalization(RevIN) and increasing input horizon on multi-channel on the LTSF model performances over multiple in-depth experiments.

**Weaknesses:**

1. The findings of the paper in the abstract are not evidently proved in the experiments.
2. This being an investigation paper, the experiments around affine mapping don’t include Mean
Bias Error for evaluation which effectively depicts the efficacy of affine transformation.
3. The conclusion states that the affine mapping dominates in “some” LTSF models but there is no
detailed information on this generalized statement.
4. TYPOS: It’s a well-written paper. I had to point out just one on a high-level overview. Use of “Instead” and “Apart from” with “However” in the Introduction were slightly confusing. Please consider checking those.

**Questions:**

The work in this paper is detailed and had observations from extensive experiments on both simulated and real-world datasets, but the findings are inconsistent. The findings stated in the abstract are not derived across the experiments through the conclusion.
I believe there are many studies on why and how affine transformations can improve time-series forecasting models. As an investigation paper, I would recommend to consider the pros and cons of the current research to effectively conclude the findings and future work that is required on the topic. For example, in this paper, X. M. Chen, Y. Li, R. Z. Wang; Performance study of affine transformation and the advanced clear-sky model to improve intra-day solar forecasts. J. Renewable Sustainable Energy 1 July 2020; 12 (4): 043703., the results about affine transformation is similar to the findings of our paper. This paper should consider investigating the use-case used in this paper and fetch out the results and challenges faced in this use-case.

---

> ### Author Response · Authors · 2023-11-23
> **Response to Reviewer uhGz (1/2)**
>
> We appreciate the time of the reviewer and thank the reviewer for acknowleding that our paper makes solid observations and gives a detailed analysis. We provide the detailed response to each question in the following.
>
> > **Q1: However, there are no information about what affine mapping is or how affine mapping is effectively introduced in the LTSF models to improve performance.**
>
> The affine function is clearly defined in this paper, plese see the footnote of Page 1 (i.e., "In general, a linear layer is defined as an affine transformation $\mathbf{Y}=\mathbf{X}\mathbf{W}^\top+\mathbf{b}$") and Equation (1) of Section 3.1 (with a title ROLES OF AFFINE MAPPING IN FORECASTING).
>
> > **W1: The findings of the paper in the abstract are not evidently proved in the experiments.**
>
> We are unclear to the logic why the reviewer draws this conclusion since the reviewer acknowledges that "this paper gives a detailed analysis of various use-cases of LTSF on public datasets and makes solid observations."
>
> To make it clear, we relist our findings stated in the abstract here again: 1) affine mapping in some LTSF models dominates forecasting performance across commonly utilized benchmarks; 2) affine mapping can effectively capture periodic patterns but encounter challenges when predicting non-periodic signals or time series with different periods across channels; and 3) using reversible normalization and increasing input horizon can significantly enhance the robustness of models. It is not hard to find through this paper that Finding 1) is supported by Section 2, for example in Figures 2, 3 and Table 1; Finding 2) is supported by Section 3.1 and Section 4, in which we have provided extensive empirical and theoretical study of linear models on LTSF tasks; and Finding 3) is supported by Section 3.2 and Section 4, as shown in Table 3 and Figures 7, 8, 9, 12, 13.
>
> Moreover our findings are supported by all the other three reviewers (SH4h, TkUX, NWqo). The evidences are as follows:
> * Reviewer SH4h "This work provides theoretical and empirical evidence to support its findings."
> * Reviewer TkUX: "This paper offers a timely and rigorous critique of the recent trend of using transformer-based models for long-term time series forecasting, and arrives at several surprising and important empirical conclusions."
> * Reviewer NWqo: "This paper provides various types of visualization to support their claims and to present the results of experiments."
>
> Therefore, we sincrely urge the reviewer to carefully read our paper and other reviews.

---

> ### Author Response · Authors · 2023-11-23
> **Response to Reviewer uhGz (2/2)**
>
> > **Q2: As an investigation paper, I would recommend to consider the pros and cons of the current research to effectively conclude the findings and future work that is required on the topic. For example, in this paper, X. M. Chen, Y. Li, R. Z. Wang; Performance study of affine transformation and the advanced clear-sky model to improve intra-day solar forecasts. J. Renewable Sustainable Energy 1 July 2020; 12 (4): 043703., the results about affine transformation is similar to the findings of our paper. This paper should consider investigating the use-case used in this paper and fetch out the results and challenges faced in this use-case.**
>
> We resectively disagree with the reviewer on "This paper should consider investigating the use-case used in this paper and fetch out the results and challenges faced in this use-case." The reasons are as follows.
>
> The goal of this paper, as stated in the abstract and the introduction, is to provide an analysis on "previous studies have demonstrated that a single linear layer can achieve competitive forecasting performance compared to other complex architectures." As our research objects are other complicated architectures which are published in recent years, e.g., SCINet (NIPS 2022), TimesNet (ICLR 2023), DLinear (AAAI 2023), PatchTST (ICLR 2023), etc.
>
> The paper by Chen et al. (2020) published in the Renewable Sustainable Energy journal does not provide an opportunity to investigate the pros and cons of recent Long-Term Traffic Speed Forecasting (LTSF) benchmarks. As a result, it is not relevant to our work. It is important to note that not investigating this paper does not impact the contribution of our paper to the LTSF community.
>
> Moreover, in our paper, the affine functions are implemented using a linear layer as $\mathbf{Y}=\mathbf{X}\mathbf{W}^\top+\mathbf{b}$, as mentioned in the previous response. On the other hand, in the paper by Chen et al. (2020), the affine function is defined as $G_{\text{past}}=aG_{\text{est,past}}+b$, where $a$ and $b$ are scalar factors to control the bias. This definition is more similar to quantile mapping, as also mentioned in Chen et al. (2020). Besides, Chen et al. (2020) only provide experimental study on improving intra-day solar forecasts, while our study covers more fields and provides additional empirical and theoretical evidence.
>
> Additionally, our paper has received positive feedback from Reviewer TkUX, who states,  "This paper offers a timely and rigorous critique of the recent trend of using transformer-based models for long-term time series forecasting. " Reviewer SH4h also praises our paper that "The authors study the fundamental mechanisms that affect the performance of recent LTSF models, which is important for the community.
>
> Therefore, we are concerned that there may be a misunderstanding between our study and yours. We hope that these responses address your concerns. Thank you again for your feedback.

---

### Official Review · Reviewer_NWqo · 2023-11-01

**Soundness:** 2 fair
**Presentation:** 1 poor
**Contribution:** 2 fair
**Rating:** 5
**Confidence:** 4

**Summary:**

Long-term time series forecasting (LTSF) is an important problem. Based on a finding from previous work that a linear layer can achieve forecasting performance comparable to complex models such as Transformers, in this paper, the authors study the effectiveness of recent approaches and provide various findings about their limitations. The authors provide theoretical and experimental explanations to support their findings.

**Strengths:**

1. This paper provides various types of visualization to support their claims and to present the results of experiments.
2. Simplicity is always appreciated. It is nice to observe simple models can achieve performance similar to or even better than complicated models.

**Weaknesses:**

1. The main claim of Section 2 is unclear. Are simple models always better than complex models for LTSF, or is it the case only to the specific framework shown in Figure 1? What if we adopt a different but still complicated framework, such as 1-dimensional CNNs? How do the results change if we include traditional models such as AR (autoregression) or ARIMA?
2. This paper is not self-contained. The experiments in Section 2 play a crucial role to motivate this work, but there is not enough description about the models and experimental setup. For example, RevIN is mentioned several times throughout the paper, but there is no definition of it. If the page limit is a problem, the authors could have added the details to Appendix.
3. Theorem 1 and 2, which are the main theoretical contributions of this work, seem trivial. If a time series can be clearly (and linearly) separated into seasonality and trend parts, I think it is obvious that a linear layer (or any linear function) is able to learn such separation.

**Questions:**

1. What is the main improvement of this work from (Zeng et al., 2022)? This works seems like a re-verification of the claim made by the previous work.
2. What do the authors suggest as a result of this work? Should we replace the benchmarks for LTSF with simple linear models?

---

> ### Author Response · Authors · 2023-11-23
> **Response to Reviewer NWqo (1/2)**
>
> Thank you for taking the time to review our paper and for your valuable comments. We would like to address your concerns as follows.
>
> > **Q1: Are simple models are always better than complex models for LTSF, or is it only relative to the specific framework shown in Figure 1?**
>
> Thank you for your insightful question. We'd like to clarify that our primary claim is not that simple models are always better than complex models in forecasting tasks. Rather, we argue that in the context of LTSF benchmarks, which are widely used in current LTSF research, the simple models we've discussed typically demonstrate robust performance. It is worth noting that Transformer-based methods such as Informer, Autoformer, and FEDformer were once predominant in LTSF tasks. However, DLinear (Zeng et. al, 2022) showed that a simple linear layer can outperform these more complex approaches, which indicates the significance of directly modeling the relationship between input and output time series. Consequently, a trend has emerged to move away from the encoder-decoder structure towards designing modules that capture temporal and channel dependencies. These methods generally use a single projection layer to generate the final forecasting results, largely aligning with the framework we simplified in Figure 1. This is why we've chosen to analyze the efficacy of each component in these methods. We discovered the importance of RevIN and projection layer, which were not discussed in detail in the original paper.
>
> > **Q2: How would the results change with a different, yet complex framework such as 1-dimensional CNNs?**
>
> We have provided one typical CNN-based model, SCINet, specifically designed for LTSF tasks. SCINet employs a tree-like structure of 1D-CNNs to handle interleaved subsequences, downsampled from the original time series. The experimental results obtained from SCINet align well with those from other models. Notably, due to the equivalence of a single 1x1 convolutional layer and a linear layer, substituting the linear layer with a 1x1 convolutional layer will obtain the same results.
>
> > **Q3: How would the results change if traditional models such as AR or ARIMA were included?**
>
> Thank you for your suggestion to incorporate traditional models such as AR and ARIMA. However, as extensively discussed in previous works like DLinear (Zeng et al., 2022) and Informer (Zhou et al., 2020), these models tend to underperform in LTSF tasks due to the accumulation of errors caused by regressive generation. This is the primary reason we chose not to include these traditional models in our study. We only include recent methods that utilize non-autoregressive generation, as illustrated in Figure 1.
>
> > **Q4: There is not enough description about the models and experimental setup, such as the definition of RevIN.**
>
> We appreciate your insightful comments and recognize the need for enhanced clarity in our methodology, particularly for readers who may not be as familiar with the topic. To address this, we will include comprehensive descriptions of the utilized baselines and modules such as RevIN in Appendix A. We apologize for any initial lack of detail, and we are grateful for your constructive feedback.
>
> > **Q5: What is the main improvement of this work over DLinear (Zeng et al., 2022)?**
>
> In the original paper of DLinear, the authors questioned whether transformers effective for time series forecasting, and then proposed a family of linear models which outperform Transformer-based methods across multiple LTSF datasets. However, their demonstration of the robust performance of linear models was largely experimental, lacking in-depth theoretical substantiation for why linear models excel in LTSF tasks. Our research builds upon the foundation laid by DLinear, offering comprehensive empirical and theoretical results to elucidate the efficacy of linear models in LTSF tasks. Our work also highlights their limitations and propose potential remedies.
>
> Besides, we have to emphasize that one crucial factor contributing to DLinear's success is the length of the input horizon. DLinear adopts a longer input horizon (336) compared to previous methods (96), without providing a comprehensive explanation — their reasoning was limited to 'underfitting' or 'overfitting' problems when varying the input horizon length. Subsequent works, such as PatchTST, have also adopted longer input horizons (512) to enhance their forecasting performance. Conversely, some approaches like TimesNet opt for shorter input horizons to undermine DLinear's performance in comparative studies. Our research underscores the importance of the input horizon length from both empirical and theoretical viewpoints, paving the way for more fair performance comparisons in the future.

---

> > ### Author Response · Authors · 2023-11-23
> > **Response to Reviewer NWqo (2/2)**
> >
> > > **Q6: What do the authors suggest as a result of this work? Should we replace the benchmarks for LTSF with simple linear models?**
> >
> > As a result of this work, we suggest that future research should not only consider simple models like RLinear and RMLP for comparison, but also rigorously examine the efficacy of their proposed modules — for instance, the performance improvement brought by RevIN. Considering the impact of input horizon length and the use of techniques like RevIN, we advocate for more extensive ablation studies, akin to those we conducted in Section 2. And it is crucial for future work to explicitly state the length of the input horizon adopted in their experiments.
> >
> > Furthermore, considering the limitations of simple linear models, we have grounds to question the rationale behind LTSF tasks, as they tend to heavily rely on inherent periodicity within datasets. This raises potential concerns about the datasets currently in widespread use for evaluation purposes. It is possible that these datasets, with their inherent biases, may not provide a comprehensive or fair assessment of different forecasting methodologies.
> >
> > ---
> >
> > We hope we have addressed your concerns adequately. We will amend the paper to clarify the points you raised and provide more comprehensive information where needed. Thank you again for your valuable feedback.

---

### Meta-Review · Area_Chair_5Ts1 · 2023-12-07

**Metareview:**

The paper study the role of affine mappings in models for long time-series forecasting problems and empirically show on LSTF benchmarks that affine mappings with RevIn and long input horizons are competitive with state-of-the-art transformer-based forecasting models. While the empirical finding are very relevant for practitioners, reviewers had concerns about limited technical novelty of the findings in this paper over previous (DLinear and RevIn) work, and lack of a comprehensive evaluation on more datasets. In addition, the theoretical analysis of the linear models seem quite straightforward, and does not add much to the paper. I would urge the authors to consider strengthening the paper on the empirical side by conducting a more thorough evaluation on more datasets.

**Justification For Why Not Higher Score:**

The main concern with this paper is around limited technical novelty of the empirical findings in this paper over previous (DLinear and RevIn) work

**Justification For Why Not Lower Score:**

N/A

---

### Decision · Program_Chairs · 2024-01-16

Reject